# High Rate of Non-Human Feeding by *Aedes aegypti* Reduces Zika Virus Transmission in South Texas

**DOI:** 10.3390/v12040453

**Published:** 2020-04-17

**Authors:** Mark F. Olson, Martial L. Ndeffo-Mbah, Jose G. Juarez, Selene Garcia-Luna, Estelle Martin, Monica K. Borucki, Matthias Frank, José Guillermo Estrada-Franco, Mario A. Rodríguez-Pérez, Nadia A. Fernández-Santos, Gloria de Jesús Molina-Gamboa, Santos Daniel Carmona Aguirre, Bernardita de Lourdes Reyes-Berrones, Luis Javier Cortés-De la cruz, Alejandro García-Barrientos, Raúl E. Huidobro-Guevara, Regina M. Brussolo-Ceballos, Josue Ramirez, Aaron Salazar, Luis F. Chaves, Ismael E. Badillo-Vargas, Gabriel L. Hamer

**Affiliations:** 1Department of Entomology, Texas A&M University, College Station, TX 77843, USA; markus9@tamu.edu (M.F.O.); jua05396@tamu.edu (J.G.J.); selene.marysol@gmail.com (S.G.-L.); estellemartin@ufl.edu (E.M.); ibadill1@southtexascollege.edu (I.E.B.-V.); 2Veterinary Integrative Biosciences, College of Veterinary Medicine, Texas A&M University, College Station, TX 77843, USA; mndeffo@cvm.tamu.edu; 3Biosciences and Biotechnology Division, Chemistry, Materials and Life Sciences Directorate, Lawrence Livermore National Laboratory, Livermore, CA 94550, USA; borucki2@llnl.gov (M.K.B.); frank1@llnl.gov (M.F.); 4Instituto Politécnico Nacional, Centro de Biotecnología Genómica, Cd. Reynosa 88710, Tamaulipas, Mexico; joseestradaf@hotmail.com (J.G.E.-F.); drmarodriguez@hotmail.com (M.A.R.-P.); nadiafriend@hotmail.com (N.A.F.-S.); 5Secretary of Health of the State of Tamaulipas, Epidemiology Directorate, Cd. Victoria 87000, Tamaulipas, Mexico; gloria.molina@tam.gob.mx (G.d.J.M.-G.); santos.carmona@tam.gob.mx (S.D.C.A.); lespt@hotmail.com (B.d.L.R.-B.); blanca_luis5@hotmail.com (L.J.C.-D.l.c.); alejandro.garcia@tam.gob.mx (A.G.-B.); Raul.huidobro@tam.gob.mx (R.E.H.-G.); rmbrussolo@gmail.com (R.M.B.-C.); 6Health Department, City of Harlingen, TX 78550, USA; jramirez@myharlingen.us; 7Hidalgo County Health & Human Services, Edinburg, TX 78539, USA; aaron.salazar@hchd.org; 8Instituto Costarricense de Investigación y Enseñanza en Nutrición y Salud (INCIENSA), Apartado Postal, Tres Ríos, Cartago 4-2250, Costa Rica; lfchavs@gmail.com

**Keywords:** Zika virus, *Aedes aegypti*, *Culex quinquefasciatus*, host selection, reproductive number

## Abstract

Mosquito-borne viruses are emerging or re-emerging globally, afflicting millions of people around the world. *Aedes aegypti*, the yellow fever mosquito, is the principal vector of dengue, Zika, and chikungunya viruses, and has well-established populations across tropical and subtropical urban areas of the Americas, including the southern United States. While intense arboviral epidemics have occurred in Mexico and further south in the Americas, local transmission in the United States has been minimal. Here, we study *Ae. aegypti* and *Culex quinquefasciatus* host feeding patterns and vertebrate host communities in residential environments of South Texas to identify host-utilization relative to availability. Only 31% of *Ae. aegypti* blood meals were derived from humans, while 50% were from dogs and 19% from other wild and domestic animals. In *Cx. quinquefasciatus*, 67% of blood meals were derived from chicken, 22% came from dogs, 9% from various wild avian species, and 2% from other mammals including one human, one cat, and one pig. We developed a model for the reproductive number, *R*_0_, for Zika virus (ZIKV) in South Texas relative to northern Mexico using human disease data from Tamaulipas, Mexico. We show that ZIKV *R*_0_ in South Texas communities could be greater than one if the risk of human exposure to *Ae. aegypti* bites in these communities is at least 60% that of Northern Mexico communities. The high utilization of non-human vertebrates and low risk of human exposure in South Texas diminishes the outbreak potential for human-amplified urban arboviruses transmitted by *Ae. aegypti*.

## 1. Introduction

Mosquito-borne viruses driven principally by *Aedes aegypti*, have emerged and re-emerged globally, resulting in a large burden of human disease [1]. Globalization and other anthropogenic factors have allowed this mosquito to thrive in diverse landscapes and facilitate urban transmission cycles of dengue virus (DENV), chikungunya virus (CHIKV), Zika virus (ZIKV), and others [2]. In the Americas, all four serotypes of DENV have re-emerged causing consistent epidemics from South America to Mexico and the Caribbean [3]. The Asian lineage of CHIKV first arrived in the Caribbean in 2014 and spread throughout the Americas in just a few years [4], resulting in 338,963 confirmed human cases [5]. ZIKV invaded Brazil in 2013 [6] and rapidly swept through the Americas in a similar fashion, resulting in an estimated 8.5 million cases in Brazil alone [7].

In the continental United States, *Ae. aegypti* is found throughout the southern states, and recent enhanced surveys of *Stegomyia* mosquitoes have documented the presence of this species in 26 states [8]. Despite this wide distribution of the primary vector and the Asian tiger mosquito (*Ae. albopictus*), a secondary vector for these viruses in many locations, the only regions experiencing autochthonous transmission of DENV, CHIKV, and ZIKV by mosquito exposure, are South Florida and South Texas [9,10]. While the Mexico cities along the U.S.—Mexico border have experienced consistent epidemics of *Ae. aegypti*-driven viruses, markedly fewer human cases have occurred in the communities of the Lower Rio Grande Valley (LRGV) on the Texas side of the border. For example, the state of Tamaulipas, Mexico, recorded an estimated 11,760 probable cases of DENV and 2677 cases of Dengue Hemorrhagic Fever between 2009 and 2019 [11]. In contrast, in the LRGV, local mosquito-borne DENV epidemics occurred only in 2005 and 2013 [9], and the outbreaks were associated with relatively small numbers of human cases. For example, in 2005 the LRGV documented three symptomatic cases and six asymptomatic cases of DEN with no travel history [12], compared to 7062 reported DEN cases in Tamaulipas the same year [13]. In contrast, these viruses are recorded with only isolated cases of local transmission in South Texas, including a single CHIKV case in Brownsville, TX in 2015 (Texas Department of State Health Services, 2016) and 11 cases of locally acquired ZIKV in the LRGV between 2016–2017 [14].

*Aedes aegypti*-driven viruses continue to have intense epidemics in the Americas, resulting in high rates of viremic humans entering the U.S. [15], but minimal local transmission has occurred [16]. This discrepancy in the magnitude of virus transmission along geo-political boundaries of the U.S.—Mexico border has attracted research attention to identify the mechanisms responsible for these patterns. This is especially perplexing given that *Ae. aegypti* in U.S. border communities has comparable relative abundances in residential neighborhoods to areas with a much higher burden of human disease across the border in Mexico [14,17]. Prior studies have identified several factors contributing to this discrepancy, which have identified social-ecological factors, such as window screens and air conditioning, that reduce the risk of exposure to the viruses [13,18]. However, despite evidence that housing quality is associated with virus transmission [18], there remains limited knowledge of how this influences the ability of *Ae. aegypti* to feed on humans and how proportional human feeding might drive virus transmission potential on both sides of the border.

This study quantifies *Ae. aegypti* host feeding patterns and vertebrate host availability in residential environments in South Texas, to compare the observed frequency of blood meals relative to the expected frequency in the study location. We rely on empirical data on *Ae. aegypti* abundance, human population density, and epidemiological data of epidemics in South Texas and in Tamaulipas, Mexico, to evaluate the risk of human-amplified urban arbovirus transmission. We present evidence contrary to the most commonly reported observation that *Ae. aegypti* feeds mostly on humans by showing a high utilization of dogs and other non-human hosts in South Texas, and that this contributes to a lower risk of human exposure to ZIKV, which reduces epidemic potential.

## 2. Materials and Methods

### 2.1. Study Site and Mosquito Collection

Blood-engorged mosquitoes were collected from several neighborhoods in the Lower Rio Grande Valley (LRGV) on the U.S. side of the U.S.–Mexico border (see Figure 1) from September, 2016 through December, 2018. The climate of Weslaco, Texas, which was used as a representation of the general climate for the LRGV, includes an average annual high and low temperature of Weslaco (28.7 and 17.4 °C) and Reynosa, Mexico (29.2 and 17.3 °C; climate-data.org). Average annual precipitation (in mm) is 609 for Weslaco and 532 for Reynosa. We sampled mosquitoes from eight lower-income (15,000–29,999 USD annual household income) neighborhoods (Mercedes, Mesquite (MM); Donna, Figueroa (DF); Mercedes, Chapa (MCH); Progreso Fresno/Progreso Encino (PF/PE); Indian Hills East (IHE); Indian Hills West (IHW); La Piñata (LP); Tierra Bella (TB)) and four middle-income (30,000–40,000 USD annual household income) neighborhoods (La Feria, La Bonita (LF); Mercedes, Rio Rico (MRR); McAllen, La Vista (MLV); Weslaco, Christian Court (WCT)) as described by Martin et al. [14]. We used Biogents Sentinel 2 traps (BGS2; Biogents, Germany) with BG lures at IHE, IHW, LP, TB, and DF neighborhoods, placing one trap outside homes once per week for a 24-h trapping period. We also used Autocidal Gravid Ovitraps (AGO) at PF/PE, DF, LF, MCH, MLV, MM, and MCH. One AGO was placed inside the home and an additional trap was placed outside the home, serviced weekly. More details about AGO preparation and deployment are described in Martin et al. [14]. The selection of homes was largely dictated by obtaining permission from the homeowners to place traps in and around the property. Upon collection, mosquitoes were identified morphologically based upon illustrations and dichotomous keys found in *The Illustrated Key to Common Mosquitoes of Louisiana* [19]. While processing mosquitoes to identify species and sex, all bloodfed mosquitoes were placed individually into nuclease-free, 1.5 mL micro-centrifuge tubes, labeled with species, sex, date and house identification number and stored at −20 or −80 °C until further processing.

### 2.2. Blood Meal Analysis

Mosquito samples were identified under microscope, photographed, and given a Sella score (stages of blood digestion and ovary development) based upon observation of the engorged abdomen [20]. The Sella score was used to identify the engorged mosquitoes that had the highest likelihood of yielding a DNA sequence. To minimize exogenous DNA on the mosquito exoskeleton, each whole mosquito was washed in 10% bleach followed by two rinses with nuclease-free water [21,22,23,24]. On a clean, chilled microscope slide, the abdomen was carefully separated from the rest of the mosquito body, and the abdominal contents were expressed into a new, labeled, DNA-free 1.5 mL microcentrifuge tube. A homogenizing bead and 200 µL of lysis solution were added to the tube with blood and shaken for 1 min at 30 Hz in the Qiagen Tissue Lyser (Qiagen, Germantown, MD, USA). DNA was extracted using the Thermo Scientific™ Kingfisher™ Flex Purification System, along with the MagMAX Core Nucleic Acid Purification Kit (Thermo Fisher Scientific, Waltham, MA, USA) following the manufacturer’s instructions.

We adopted previously-published protocols to conduct a PCR-Sanger sequencing blood meal analysis [25,26,27]. Three primer pairs were used in a tiered approach: (I) A vertebrate cocktail targeting a 648 base pair region of the cytochrome c oxidase 1 (COI) gene, (II) blood meal (BM) primers targeting a 358 base pair region of the cytochrome b gene, and (III) ‘Herp’ primers that target a 228 base pair region of the cytochrome b gene (Appendix A) [27]. This three-tiered approach has the benefits of cost efficiency, maximizing the number of identified blood meals, and increasing reliability of results [25,26,27]. First, every sample was tested using the vertebrate cocktail primers. Samples producing an amplicon of 648 bp were cleaned using ExoSAP-IT (Thermo Fisher Scientific) and submitted to Eton Bioscience (San Diego, CA, USA) for Sanger sequencing. Sequencing results were analyzed using Geneious R9 software (Biomatters, Ltd., Auckland, New Zealand). Only samples with ≥ 95% pairwise identity match to a vertebrate host sequence in NCBI, and ≥ 95% grade (a weighted score comprised of *e*-value, pairwise identity, and the coverage) were accepted as a confirmed result.

Based upon the outcome of the initial PCR, we continued the iterative bloodmeal analysis PCR process if there was (I) match to human basic local alignment search tool (BLAST), (II) no PCR amplicon, (III) poor sequence quality, or (IV) evidence of mixed DNA (double-nucleotide peaks in chromatograph) [27]. If we obtained any of these four outcomes, a second PCR utilizing the BM1:BM2 primers was conducted [25,27]. Finally, using the same criteria, the analysis was either concluded or subjected to a third primer pair and PCR thermal profile, the ‘Herp’ primers [25,27].

We followed the protocol of Medeiros et al. [27] for the vertebrate cocktail reaction, but modified the thermal cycling conditions as follows: after denaturation we ran nine cycles of 94 °C for 30 s, a gradient from 45 to 54 °C for 40 s, and 72 °C for 1 min. The remaining thermal cycling conditions were identical to the Medeiros protocol. For the BM and ‘Herp’ reactions, we followed the protocol of Hamer et al. [25] with the following modification: we lowered the annealing temperature for the ‘Herp’ reaction from 50 to 47 °C. The vertebrate cocktail, BM1:BM2, and ‘herp’ PCR reactions used the following reagents and quantities per reaction: 8.59 µL Nuclease-free H_2_O, 12.5 µL FailSafe™ PCR 2X Premix E (Lucigen, Middleton, WI, USA), 0.83 µL forward primer, 0.83 µL reverse primer, 0.25 µL FailSafe™ PCR Enzyme Mix (Lucigen), and 2 µL DNA template.

The protocol was tested using lab-raised, *Aedes aegypti* and *Culex quinquefasciatus* mosquitoes fed on defibrinated sheep blood (HemoStat Laboratories, Dixon, CA, USA). Adult female mosquitoes were offered artificial blood meals using a Hemotek membrane feeder (Hemotek Ltd., Blackburn, UK). Each specimen was observed under a dissecting light microscope to confirm species, give a Sella score, and capture a digital photograph. We also extracted DNA directly from the blood of several controls including iguana (*Iguana iguana*), white-tailed deer (*Odocoileus virginianus*), tiger (*Panthera tigris*), sandhill crane (*Grus canadensis*), and sheep (*Ovis aries*). These vertebrate species were selected because they are unlikely to be found in mosquito blood meals from this region and would thus minimize the risk of downstream amplicon contamination.

### 2.3. Molecular Verification of Mosquito Species

While most mosquitoes captured via BGS2 traps could be taxonomically classified from morphological features, those collected from the glue boards of the AGO traps are often damaged and more difficult to identify morphologically. Therefore, molecular identification of mosquito species was confirmed using a modified version of the protocol designed by Folmer et al. [28]. Briefly, a primer pair that amplifies a 710-bp fragment of the cytochrome c oxidase subunit I gene (LCO 1490 and HCO 2198; Appendix A) was used with the following reagents and quantities per reaction: 8 µL Nuclease-free H_2_O, 12.5 µL FailSafe™ PCR 2X Premix E (Lucigen), 1 µL forward primer (LCO 1490), 1 µL reverse primer (HCO 2198), 0.5 µL FailSafe™ PCR Enzyme Mix (Lucigen), and 2 µL DNA template. The PCR thermal cycling profile included initial denaturation for three minutes at 95 °C followed by 35 cycles of 95 °C for 1 min, 45 °C for 1.5 min, and 72 °C for 2 min, followed by a final extension at 72 °C for 5 min. Amplified PCR products were purified using Exo-SAP-IT™ (ThermoFisher Scientific) and sent to Eton Bioscience (San Diego) for Sanger sequencing.

### 2.4. Quantitative Synthesis of Published Literature

We compiled all the published data on *Ae. aegypti* host feeding patterns from around the world. To systematically review the literature, we searched PubMed, Web of Science, and Google Scholar for published literature using keywords “*Aedes aegypti* host feeding”, and a second search of “*Aedes aegypti* blood meal analysis”. These queries in Web of Science yielded 50 and seven results, respectively, in PubMed yielded 13 and four results, respectively, and in Google Scholar yielded 1970 and 450 results, respectively. We also tracked references from key review papers and other primary literature. Inclusion criteria included blood meal results from wild-caught mosquitoes. Studies using laboratory-raised mosquitoes, as well as studies which indicated samples were likely from the form *Ae. aegypti formosus* [29,30,31], were excluded. The form *formosus* was excluded to allow a focus on the urban form of *Ae. Aegypti*, which is globally distributed.

### 2.5. Mosquito Relative Abundance

Female *Ae. aegypti* relative abundance was estimated in the LRGV and the city of Reynosa, Tamaulipas, Mexico, using AGO traps that were deployed concurrently on both sides of the border in 2017 (Figure 1). Eighty AGO traps were deployed outside residential homes in Reynosa and checked weekly between May 7 and Aug 12 (trapping data were unavailable for two weeks in this period due to adverse weather or trap failure) [32]. In the LRGV, 30 AGO traps were deployed outside residential homes during the same weeks as a subset of the data presented in a previous study [14]. 

### 2.6. Vertebrate Surveys

In order to estimate mosquito host selection, a questionnaire related to vertebrate availability was developed and conducted in the four primary communities where blood-fed mosquitoes were collected. Project personnel visited all the homes containing BG Sentinel 2 traps in these communities: 14 (out of 307 total homes present) in Indian Hills East, 10 (96) in Indian Hills West, 13 (160) in La Piñata, and seven (49) in Tierra Bella. An adult from each home was asked for the number of persons living at each residence (further categorized by age group < 5; 5–17; 18–65; > 65), the number of dogs, cats, pet birds, chickens, pigs, horses, and other animals. Of the dogs and cats, the number of them roaming outside of the property was also noted. From previous observations, we suspected that a large number of stray dogs and cats live in some of these neighborhoods, therefore a final question regarding the number of strays that the adult resident is aware of was also asked. Results from the surveys were tabulated and relative abundance calculated with 95% confidence intervals for selected vertebrates, using the Wilson/Brown method (Appendix A) [33]. 

Populations of human, dog, cat, chicken, pig, and opossum were estimated by extrapolating the vertebrates documented from survey homes to create estimates of the number of each vertebrate per unit area in the entire community. To achieve this, the average number of vertebrates in the surveyed homes was multiplied by the total number of homes in the defined community to arrive at an estimated density of each vertebrate per unit area. This number was divided by the total estimated number of all potential hosts to obtain relative abundance. Population estimates of wild birds and wild mammals (rodents, meso-predators, etc.) were not obtained. 

### 2.7. Human Density Estimation

We used remote sensing satellite imagery (Google Earth, California, USA) to map the communities within the LRGV using QGIS 3.4 (QGIS Development team 2019). We estimated household densities using the 2010 US census blocks shape file and extracted the information regarding number of houses, number people/house and area of the community. We also quantified household density in Nuevo Amanecer as a representative community in the city of Reynosa with prior DENV transmission activity (Rodríguez-Pérez Mario A, A. M. A., Russell Tanya L, Olguin-Rodriguez Omar, Laredo-Tiscareño Stephanie V, Garza-Hernandez Javier A, Reyes-Villanueva Filiberto. Host-seeking Aedes aegypti linked to dengue seropositive households at northeastern Mexico. Journal of Vector Borne Diseases (in press)). We used satellite imagery (Google Earth, California, USA) to quantify homes manually and census block information to identify the boundaries of the neighborhood. Nuevo Amanecer was chosen because of its known dengue endemicity.

### 2.8. Host Selection Indices

We estimated the Forage Ratio (FR), the frequency at which a mosquito selects a vertebrate host over other available vertebrate hosts, by dividing the observed frequency of bloodmeals divided by the expected frequency of bloodmeals of a given species [34]
FR = *s*/*a*,(1)
where *s* = the percent of female mosquitoes containing blood of a particular host, and *a* = percent of the total available host population represented by that particular host [35]. A forage ratio of 1.0 indicates mosquitoes are feeding on hosts in equal proportion to availability, whereas values >1.0 indicate over-utilization and values <1.0 indicate under-utilization. We used the Wilson/Brown statistical method to calculate 95% confidence intervals [33].

We also estimated the human blood index (HBI), which measures the frequency at which female mosquitoes feed on human hosts and is the number of human blood meals divided by the number of engorged females [36].

### 2.9. Tamaulipas Human Disease Data

The General Directorate of Epidemiology, Secretariat of Health, México aggregates probable and confirmed empirical cases of DEN, CHIK, and ZIK in the state of Tamaulipas (Appendix A). Patients with history of travel outside of Tamaulipas in the month prior to onset of symptoms were not included in the modeling of $R_{0}$. Physicians of symptomatic patients use a case definition of DEN: fever with ≥2 signs or symptoms such as retro-orbital or ocular pain, rash, headache, arthralgia, myalgia, leukopenia or hemorrhagic manifestations; CHIK: severe arthralgia, intense asymmetric, debilitating joint pain, swelling associated with tenosynovitis; and, ZIK: pruritic maculopapular rash, for differential clinical diagnosis between the three viruses and are required to report cases to the Secretary of Health of Tamaulipas. Clinical serum samples receiving laboratory confirmation were sent by sanitary jurisdictions of the state to the Molecular Biology Laboratory of the Tamaulipas State Public Health Laboratory. They were stored at −20 °C until further processing.

Nucleic acid extraction was performed using a MagNA Pure LC total nucleic acid isolation kit in a MagNA Pure LC 2.0 Instrument (Roche Applied Science, Germany). The extracted viral RNA was stored at −70 °C. We used RT-PCR to detect the presence of arboviruses using protocols previously described [37]. We used the SuperScript III Platinum^®^ One-Step qRT-PCR System enzyme (Invitrogen, Carlsbad, CA, USA). A 7500 Fast Real-Time Thermocycler from Applied Biosystems (Foster City, CA, USA) was used, and reportable positive values were below a *C*t value of 38.

For ZIKV, the primary patients that received laboratory confirmation using RT-PCR were pregnant females. For the modeling in this study, we used the empirical data of probable cases of ZIKV in the municipality of Reynosa in 2017 with no recent travel history (Appendix A).

### 2.10. Mathematical Modeling

The Ross MacDonald formulation of the basic reproductive number for mosquito-borne diseases [38] is defined as the average number of secondary human cases generated by an index case in an otherwise susceptible population
(2)$$R_{0}=\frac{m{\left(af\right)}^{2}bce{\;}^{-\mu EIP}}{\mu \;r},$$
where *m*: the density of female mosquitoes to human, *a*: female mosquito biting rate, *f*: the proportion of mosquito feeding on human, *b*: mosquito-to-human transmission probability, *c*: human-to-mosquito transmission probability, *EIP*: extrinsic incubation period, 1/*r*: human average infectious period, μ: adult mosquito mortality rate [38]. The density of *Ae. aegypti* to humans is a function of mosquito density and human exposure to mosquitoes [38,39,40] *m =* D×E where D is mosquito density and E is the human risk of exposure to *Ae. aegypti*. The risk of exposure to mosquitoes is a function of socioeconomic variables, such as the availability of air conditioning, which can drastically limit mosquito–human contacts and virus transmission, even when mosquitoes are abundant [18]. Studies have shown that population mobility may also play a role in individuals exposure risk to *Ae. aegypti* [41]. Furthermore, fine-scale variation in population susceptibility, immunity, or social structures may also be factors contributing to vector-borne disease transmission heterogeneity amongst neighboring communities [42]. Though these parameters may be hard to measure empirically, they can play a pivotal role in the risk of mosquito borne disease outbreaks.

Given the geographical proximity and similarities between the city of Reynosa and neighborhoods in the LRGV of South Texas, any difference in the risk of outbreak $(R_{0}$), for a newly introduced *Ae. Aegypti*-borne disease such as Zika, between the two communities would be due to the density of mosquito to human (*m*) or the proportion of mosquito feeding on humans (*f*). Therefore, $R_{0}$ in the LRGV and Reynosa can be written as $R_{0}^{LRGV}=\frac{m_{LRGV}{\left(af_{LRGV}\right)}^{2}bce{\;}^{-\mu EIP}}{\mu \;r}$ and $R_{0}^{Rey}=\frac{m_{Rey}{\left(af_{Rey}\right)}^{2}bce^{-\mu EIP}}{\mu \;r}$, respectively. We have $R_{0}^{Rey}=m_{Rey}{\left(f_{Rey}\right)}^{2}\frac{{\left(a\right)}^{2}bce^{-\mu EIP}}{\mu \;r}$, which is rearranged as $\frac{{\left(a\right)}^{2}bce{\;}^{-\mu EIP}}{\mu \;r}=R_{0}^{Rey}\frac{1}{m_{Rey}{\left(f_{Rey}\right)}^{2}}$ This implies that
$$R_{0}^{LRGV}=m_{LRGV}{\left(f_{LRGV}\right)}^{2}\frac{{\left(a\right)}^{2}bce{\;}^{-\mu EIP}}{\mu \;r}=m_{LRGV}{\left(f_{LRGV}\right)}^{2}R_{0}^{Rey}\frac{1}{m_{Rey}{\left(f_{Rey}\right)}^{2}}$$
(3)$$R_{0}^{LRGV}=R_{0}^{Rey}\frac{m_{LRGV}}{m_{Rey}}{\left(\frac{f_{LRGV}}{f_{Rey}}\right)}^{2},$$
where $R_{0}^{LRGV}$ and $R_{0}^{Rey}$ are the basic reproductive numbers in the LRGV and Reynosa, respectively. $m_{LRGV}$ and $m_{Rey}$ are the density of female *Ae. aegypti* to human in the LRGV and Reynosa, respectively; and $f_{LRGV}$ and $f_{Rey}$ are the proportion of *Ae. aegypti* feeding on humans in the LRGV and Reynosa, respectively. We estimated the ratio $\frac{m_{LRGV}}{m_{Rey}}=\frac{D_{LRGV}E_{LRGV}}{D_{Rey}E_{Rey}}$. Female *Ae. aegypti* relative abundance was estimated using AGO data collected in 2017 during the same weeks in the city of Reynosa (5.16 female *Ae. aegypti* per AGO per week) and in the LRGV (4.16 female *Ae. aegypti* per AGO per week) [14]. So $\frac{m_{LRGV}}{m_{Rey}}=0.8\frac{E_{LRGV}}{E_{Rey}}$. As the proportion of feeding on human in Reynosa is not currently available, we considered a range of values informed by available data from the Americas (Table 1): LRGV, Puerto Rico, and Florida.

$R_{0}$ in Reynosa municipality was estimated using case data for the 2017 Zika epidemic and the EstimateR function from the EpiEstim R library [43,44] to estimate the time-dependent reproductive number, R(t), based on the method introduced by Cori et al. [43]. We derive $R_{0}$ using the fact that $R_{0}$ is equal to R(t) at the start of the outbreak, such as Zika, for which we do not have pre-existing immunity in the population [44]. This approach would not be applicable to endemic diseases such as dengue. The instantaneous reproduction number R(t) was computed over 4-week sliding windows using the method introduced by Cori et al. [43]. This approach uses a Bayesian inference method to propagate uncertainty of data and generation time into $R_{0}$ estimate. Following Ferguson et al. [44], we assume that the ZIKV generation time is gamma-distributed with a mean of 20.0 days and a standard deviation (s.d.) of 7.4 days. The incidence data themselves may contain many potential sources of uncertainty such as misdiagnosis, variable time-dependent case detection rate, and asymptomatic cases, which are not explicitly taken into account into our analysis.

**Table 1 viruses-12-00453-t001:** Published studies of *Ae. aegypti* host feeding patterns.

				Feeding Patterns on Vertebrates (%)							
Citation	Location	Method ^a^	Site ^b^	Human	Mix/Human	Dog	Cat	Other Mammal	Avian	Unknown	Total
[45]	Nigeria	Ab	In/Out	7 (44%)					1 (6%)	8 (50%)	16
[46]	Tanzania	Ab	In	45 (100%)							45
[47]	Kenya—coast	Ab	In/Out	165 (94%)		1 (0.5%)	1 (0.5%)	9 (5%)			176
[48]	South Africa	Ab	Out	3 (75%)						1 (25%)	4
[49]	India, Poona	Ab	In	17 (81%)						4 (19%)	21
[50]	India	Ab	In	49 (96%)						2 (4%)	51
[49]	Malaya	Ab	In	109 (99%)		1 (1%)					110
[51]	Hawaii	Ab	Out	339 (54%)		117 (19%)	21 (3%)	71 (11%)	3 (0.5%)	80 (13%)	631
[52]	Thailand	Ab	In/Out	789 (88%)	66 (7.4%)	2 (2.2%)	4 (0.5%)	8 (1%)	9 (1%)		896
[53]	Puerto Rico	Ab	In	1483 (95%)	31 (2%)	47 (3%)					1561
[54]	Thailand—single host	Ab	In/Out	658 (99%)			1	4 (0.6%)	1		664
[54]	Thailand—mixed	Ab	In/Out		86 (98%)						88
[55]	E. Australia	DNA	Out	131 (75%)	7 (4%)	23 (13%)	2 (1%)	1 (0.5%)	10 (6%)		174
[56]	Thailand	DNA	N/A	766 (86.1%)	32 (3.6%) *	18 (2%)		39 (4.4%)		35 (3.9%)	890
[57]	Puerto Rico-P	DNA	Out	101 (76.2%)		27 (20.8%)	3 (2.3%)	1 (0.8%)			132
[57]	Puerto Rico-R	DNA	Out	210 (78.9%)	1 (0.4%)	49 (18.4%)	3 (1.1%)		3 (1.1%)		266
[58]	India	Gel precip	In/Out	129 (87.8%)				11 (7.5%)	1 (0.7%)	6 (4%)	147
[59]	India	Gel precip	Out	54 (96.4%)				2 (3.6%)			56
[60]	Mexico	DNA	In/Out	223 (98%)						5 (2%)	228
[61]	Florida—IR	DNA	Out	111 (90.2%)				11 (8.9%)	1 (0.8%)		123
[61]	Florida—M	DNA	Out	8 (61.5%)				5 (38.5%)			13
[62]	Grenada	DNA	Out	22 (70%)		2 (6%)	1 (3%)	6 (18%)	1 (3%)		32

^a^ Ab = precipitin test for presence of antibody, DNA = molecular identification, Gel precip = agarose gel precipitin technique. ^b^ Indoor = In, Outdoor = Out. * Samples were positive for two hosts, but the authors did not reveal which two hosts. It is assumed that one of the hosts is human.

## 3. Results

### 3.1. Blood Meal Analysis

In total, 230 bloodfed *Ae. aegypti* (Sella score of 2–5) [20] were collected, molecularly confirmed to species and processed for the blood meal analysis (four indoor, 226 outdoor; 181 using BGS2 traps, 49 using AGO). Of these, 186 (81%) yielded a bloodmeal analysis result which include 50% (*n* = 93) from dogs (*Canis lupus familiaris*), 31% (*n* = 57) from humans (*Homo sapiens*), 12% (*n* = 22) from cats (*Felis catus*), 3% (*n* = 6) from chicken (*Gallus gallus*) and 4% from other mammals (Table 2). Of the four *Ae. aegypti* collected indoors by AGO traps, three yielded a result (one human, two dogs). Bloodfed *Ae. aegypti* with results came from two different homes in MM, two homes in DF, one home in MCH, four homes in PF/PE, 34 homes in IHE, nine homes in IHW, 19 homes in LP, 10 homes in TB, one home in LF, two homes in MLV, and two homes in WCT. For *Cx. quinquefasciatus*, 124 bloodfed individuals (Sella score of 2–4) were collected, molecularly confirmed to species, and processed for bloodmeal analysis (0 indoor, 124 outdoor; 113 using BGS2 traps, 11 using AGO traps). Of these, 123 (99%) yielded a bloodmeal analysis result which included 67% (*n* = 82) from chicken, 22% (*n* = 27) were from dog, 9% (*n* = 11) from six wild bird species and 2% from other mammals (Table 3). Two *Ae. aegypti* samples had mixed bloodmeals including dog and human, while no *Cx. quinquefasciatus* had evidence of mixed bloodmeals. The success of the vertebrate host identification of the blooded abdomen for *Ae. aegypti* was significantly different across Sella scores (*p* = 0.0333; 92% for Sella score of two, 76% for three, 33% for four, and 25% for five). The success of the vertebrate host identification of the blooded abdomen for *Cx. quinquefasciatus* was not significantly different among Sella scores (*p* = 0.3333; 99% for Sella score of two, 100% for three, and 100% for four). The quantitative analysis of 18 published studies of *Ae. aegypti* host feeding patterns reveals that humans are the dominant host with an average of 83.1% (Table 1). If we only consider prior studies with outdoor mosquito collections, the average percentage of human feeding is 85%. Only two studies, one in Nigeria (Table 1) and this current study from South Texas, reveal feeding patterns where humans represent less than half of the bloodmeals. 

### 3.2. Mosquito Relative Abundance

Mosquito sampling between May 7 to August 13, 2017 using 80 Sentinel AGO traps in Reynosa yielded an average of 5.16 (± 0.43 SEM) female *Ae. aegypti* per AGO per week (Appendix A). In the LRGV, 30 Sentinel AGO traps during these same weeks yielded an average of 4.16 (± 0.43 SEM) female *Ae. aegypti* per AGO per week [14]. 

### 3.3. Vertebrate Surveys and Population Density

We conducted a vertebrate questionnaire for 44 homes in four communities asking about all vertebrates living in the home, property, or neighborhood (Appendix A). The average number of occupants per home was 4.7 (± 0.41 SEM) and the total estimated number of homes in all four communities was 612. With our vertebrate surveys, we estimated 5,146 humans per km^2^, 4,161 dogs per km^2^, 1,751 cats per km^2^, 2,299 chickens per km^2^, and 75 pigs per km^2^ (Appendix A). Independent from the household questionnaires, our analysis of US Census data using QGIS for the combined communities in the current study where blood-fed individuals were collected and in the eight communities with AGO surveillance [14], we estimate that on average the human density was 3,597 per km^2^ (Appendix A). In Nuevo Amanecer, Reynosa, we identified 885 homes in the neighborhood minus the soccer field open space. The area with homes is 0.27 km^2^ and using an average occupancy of 4.2 persons per home (based on unpublished data from co-author M. A. Rodríguez-Pérez), the estimated human density for this area is 13,767 per km^2^. The human density in Reynosa is between 2.7- and 3.8-fold higher than comparable low-income communities in the LRGV.

### 3.4. Host Selection

Forage ratios for *Ae. aegypti* and *Cx. quinquefasciatus* were calculated with host availability estimated from our vertebrate surveys in the neighborhoods where we collected engorged mosquitoes. The *Ae. aegypti* forage ratio (observed frequency of bloodmeals from a given host divided by the expected frequency) for dogs (1.61) was nearly twice the forage ratio for humans (0.81; Table 2). In contrast, the highest forage ratio for *Cx. quinquefasciatus* was on chicken (3.92; Table 3). The human blood index (total number of human blood meals divided by the total number of engorged females with a confirmed result) for *Ae. aegypti* and *Cx. quinquefasciatus* was 30.7% and 0.8% respectively.

### 3.5. Mathematical Modeling

Using the Ross MacDonald equation for the basic reproductive number, $R_{0}$, and based on the 2017 cases of ZIKV in Reynosa (Appendix A) and data on *Ae. aegypti* collected in the LRGV and Reynosa, we estimated ZIKV $R_{0}^{LRGV}$ in the LRGV. We started by estimating $R_{0}^{Rey}$ in Reynosa using case data from the 2017 Zika outbreak in Reynosa, where $R_{0}$ was 2.2 (95% Confidence Interval: 1.1—3.8) (Appendix A). A total of 330 cases of Zika were observed in Reynosa in 2017; only a subset were tested by PCR, of which 81 were confirmed positive for ZIKV RNA. Seven Zika cases were not included, given a history of travel in the prior month. Because of the geographical proximity between Reynosa and the LRGV, we assumed that all parameters, except for mosquito abundance and human biting rates, in the Ross MacDonald *R*_0_ equation were equal across the US-Mexico border. We obtained the following expression for $R_{0}$ in the LRGV
(4)$$R_{0}^{LRGV}=R_{0}^{Rey}0.8\frac{E_{LRGV}}{E_{Rey}}{\left(\frac{0.31}{f_{Rey}}\right)}^{2}$$
where 0.8 is the ratio between mosquito abundance in LRGV and Reynosa estimated with AGO traps, and 0.31 is the proportion of *Ae. aegypti* feeding on humans in LRGV. $f_{Rey}$ is the proportion of *Ae. aegypti* feeding on human in Reynosa, and $E_{Rey}\;\left(E_{LRGV}\right)$ is the risk of human exposure to *Ae. aegypti* bites in Reynosa (LRGV). Then, with the $R_{0}^{Rey}$ estimate and using different combinations for the unknown parameters, *f_Rey_* and *E_Rey_*/*E_LRGV_*, of Equation (4) we studied the conditions for the establishment, i.e., *R*_0_ > 1, of a ZIKV epidemic in the LRGV (Figure 2). Our analysis shows that if *f_Rey_* = *f_LRGV_* then ZIKV outbreaks would not occur in the LRGV when the risk of human exposure to *Ae. aegypti* bites in the LRGV is below 60% of the risk in Reynosa (Figure 2). Below 60%, there are limited scenarios where Zika outbreaks may occur in the LRGV: for example, when the *f_Rey_* is smaller than *f_LRGV_* and the human exposure risk to *Ae. aegypti* bites in Reynosa is fives time larger than in LRGV (Figure 2).

## 4. Discussion

This study documents *Ae. aegypti* feeding on humans only 31% of the time in the sampled communities in South Texas; instead, the majority of bloodmeals (50%) were from domestic dogs. This is an unexpectedly low rate of human feeding given that this species is ubiquitously classified as an anthropophilic species [18,63]. The quantitative synthesis of 18 published blood meal analysis studies on *Ae. aegypti* shows that the average percent of human blood-feeding was 83.8%. In 1967, MacDonald pointed out that “Although *Ae. aegypti* has been the study of a very large number of papers, there are only a few records of its host preference [49].” A half century later, this observation remains the same, given that only 21 studies have published *Ae. aegypti* host feeding patterns, three of which concerned the subspecies *formosus*, while our review of the published literature identified 86 primary publications that have reported host feeding patterns for members of the *Cx. pipiens* complex. The less attention to *Ae. aegypti* host feeding is likely due to the assumption that this species is largely anthropophilic and reluctance to conduct the expensive bloodmeal analysis for confirmation. Zooprophylaxis is the concept that the presence of incompetent hosts can ‘waste’ bites from vector species and reduce the transmission of an infectious agent [64]. Furthermore, prior studies have identified that arboviral transmission potential is impacted by host community composition and competence [65,66]. For human-amplified urban arboviruses like ZIKV, less feeding on humans and more feeding on non-competent hosts (vertebrates with a low duration and magnitude of viremia unable to re-infect *Ae. aegypti*), will have a zooprophylactic effect on transmission, as originally observed with cattle when describing zooprophylaxis in malaria transmission [67,68]. This has principally been considered an important phenomenon in human malaria transmission with *Anopheles* spp. wasting bites on cattle, something that protects humans from malaria infectious bites [69,70,71]. However, Hess and Hayes [64] determined that potential for zooprophylaxis exists in *Cx. tarsalis, Cx. pipiens, Cx. quinquefasciatus* and *Ae. albopictus*. Moreover, in East Africa the observation of non-human host feeding by *Ae. aegypti* led to the conclusion that these were likely to be poor vectors of yellow fever virus [29].

Non-human feeding by *Ae. aegypti* may have important consequences for arboviral pathogen transmission. For example, the receptivity of certain regions of the world to *Ae. Aegypti*-driven arboviruses might vary not simply due to the abundance of *Ae. aegypti*, but due to the proclivity and availability of *Ae. aegypti* to feed on humans relative to other hosts. In that sense, threshold indices, such as R_0_, that indicate when the transmission of viruses will persist, can guide management activities and even inform urban planning and home modification to further reduce the probability of *Ae. aegypti* feeding on humans.

Collection technique and location can influence the apparent host feeding patterns. While only 2% of the collections in this study were from inside homes, all of the specimens were collected within the residential yard. While our previous study documented *Ae. aegypti* inside homes of the LRGV [14], we did not target indoor collections with aspirators, given the unique socio-demographics and political climate of the LRGV, which makes indoor access challenging. Prior studies on *Ae. aegypti* host feeding patterns are limited, with only ten studies (56%) with indoor collections, seven (39%) with collections from residential yards, and two studies (11%) with collections in non-residential locations (Table 1). Of the published studies reporting *Ae. aegypti* blood meal results, most (69%) have been conducted in regions of the world where dengue is endemic (Table 1). This identifies a research gap, with a few studies such as this one reporting *Ae. aegypti* host feeding patterns in areas where the environment may be suitable for arboviral transmission, but risk appears to be diminished by limited access to humans, and, possibly easier access to non-human vertebrates. We also did not process blood-engorged mosquitoes collected in Reynosa in the same laboratory as the current study, which is a priority for future research.

The low rate of *Ae. aegypti* anthropophily in the current study could be explained by several reasons. The most parsimonious explanation is the higher availability of non-human hosts, limited opportunity for human biting, and lower human density [18,63]. In our Texas study sites, humans make up 41% of all domestic hosts in the four study sites combined. Our analysis does not account for wild birds and wild mammals which, if included, would further reduce the relative abundance of humans compared to non-human animals. Although many human-amplified urban arboviruses occur in densely populated settings, a study in Thailand found the largest dengue epidemics occurred in low to moderate population densities, where water storage and the production of mosquitoes is an additional factor driving transmission [72]. Another factor influencing the ability of *Ae. aegypti* to feed on humans is the integrity of the home and frequency of indoor feeding. Our recent study in nearby neighborhoods shows that the outdoor *Ae. aegypti* relative abundance is about eight times that of the indoor population [14]. Prior studies in the Texas–Mexico border region have shown that the presence of air conditioning units and larger lot size are associated with a lower probability of homeowners being exposed to DENV [13,18]. A final hypothesis explaining the low rate of *Ae. aegypti* feeding on humans is that there is a genetic basis. A genetic basis for host selection has been well documented in *Culex* spp. [73] and *Anopheles* spp. [74]. In 1967, Macdonald [49] reviewed the progress on understanding the ecology of multiple forms of *Ae. Aegypti*, including *Ae. aegypti formosus* found in east Africa which tended to be more exophilic and frequently fed on non-human hosts. More contemporary population genetics studies have confirmed that *Ae. aegypti formosus* is the ancestral form of the domesticated *Ae. aegypti aegypti*, which lives in tight association with human landscapes and is more anthropophilic [75]. The host preference of the ancestral and domesticated forms of *Ae. aegypti* in east Africa is considered to have a genetic basis [76,77]. Although the domesticated form of *Ae. aegypti* has spread around the world, Macdonald [49] postulated that, outside Africa, the plasticity of the species means the potential for non-human feeding and exophily exists, and the current study supports the ability of *Ae. aegypti* to adapt to an environment with lower availability of human hosts [63,78].

With *Ae. aegypti* feeding on non-human hosts about 70% of the time, this study highlights the potential role of *Ae. aeygyti* in contributing to enzootic transmission among animals or even bridge transmission of zoonotic agents to humans. Of the non-human bloodmeals, 50% were from domestic dogs. A recent study testing dogs from animal shelters in the LRGV (Edinburg, TX, USA) identified 20.9% of the dogs to be infected with dog heartworm, *Dirofilaria immitis* [79]. Several studies suggest *Ae. aegypti* as an efficient vector of *D. immitis* in dogs [80,81], and a study in Florida found *Ae. aegypti* infected with *D. immitis* [82]. Given that *Ae. aegypti* is the dominant mammalophilic mosquito species in low- and middle-income LRGV residential communities [14], these observations suggest that *Ae. aegypti* may play a role in *D. immitis* transmission, which warrants further research. Prior studies have also documented the potential spill-over of human-amplified urban arboviruses into wild or domestic animals [83,84], suggesting that *Ae. aegypti* could play the role of a bridge vector in this context.

The blood meal analysis results for *Cx. quinquefasciatus* yielded 75.6% of the bloodmeals from birds, with chickens being the dominant species (Table 3). These results are consistent with prior studies which show that *Culex* are principally ornithophilic [85]. Both the bloodfed *Culex* and *Aedes* were processed with the exact same protocol, and the contrasting results provide more confidence in the accuracy of the identified blood meals. The high host chicken use is consistent with *Cx. quinquefasciatus* host feeding patterns in tropical and subtropical regions [86]. The inclusion of Passerines as hosts by *Cx. quinquefasciatus* would suggest their potential role as an amplification vector for West Nile virus (WNV) in South Texas, while the observation of human feeding would suggest the potential to bridge WNV to humans. In some regions of the world, *Cx. quinquefasciatus* can be highly anthropophilic [87] and it was surprising to not find more human-derived bloodmeals. A good example of a study that analyzed both forage ratio (FR) and the human blood index (HBI) for *Cx. quinquefasciatus* mosquitoes was conducted by Garcia-Rejon et al. in Yucatan State, Mexico [35]. They found an HBI of 6.7, but the FR for humans was < 1 when compared to that of other vertebrate hosts, indicating that *Cx. quinquefasciatus* mosquitoes in this area under-utilized humans as hosts. In fact, species of the Passeriformes and Galliformes orders were the only hosts that had a FR >1 [35]. This ornithophilic pattern of *Cx. quinquefasciatus* mosquitoes was also demonstrated in College Station, Texas (95.5% blood meals on birds) [34]; and Harris County, Texas (39.1% on birds) [88].

## 5. Conclusions

In conclusion, we identify a potential mechanism explaining how ZIKV resulted in large epidemics in Reynosa, Tamaulipas, Mexico but did not result in widespread transmission in the LRGV of South Texas. The population of *Ae. aegypti* in South Texas fed on humans only 31% of the time, which is likely due to the abundance of non-human hosts in the residential neighborhood, the low human density, and social practices of minimizing risk of exposure to *Ae. aegypti* [18]. The high rate of non-human blood meals of *Ae. aegypti* occurring in the LRGV is likely reducing the risk of human-amplified urban arboviruses such as DENV, ZIKV, and CHIKV. However, the high number of blood meals from dogs and cats is concerning for zoonotic agents such as dog heartworm transmission and the potential for bridge transmission to human populations [89]. The population of *Cx. quinquefasciatus* in the LRGV was ornithophilic, which likely contributes to the local transmission of WNV observed in the region. This study revealed high non-human host utilization in *Ae. aegypti* mosquitoes, which warrants further research to determine factors driving the variation in mosquito–human contact.

## Figures and Tables

**Figure 1 viruses-12-00453-f001:**
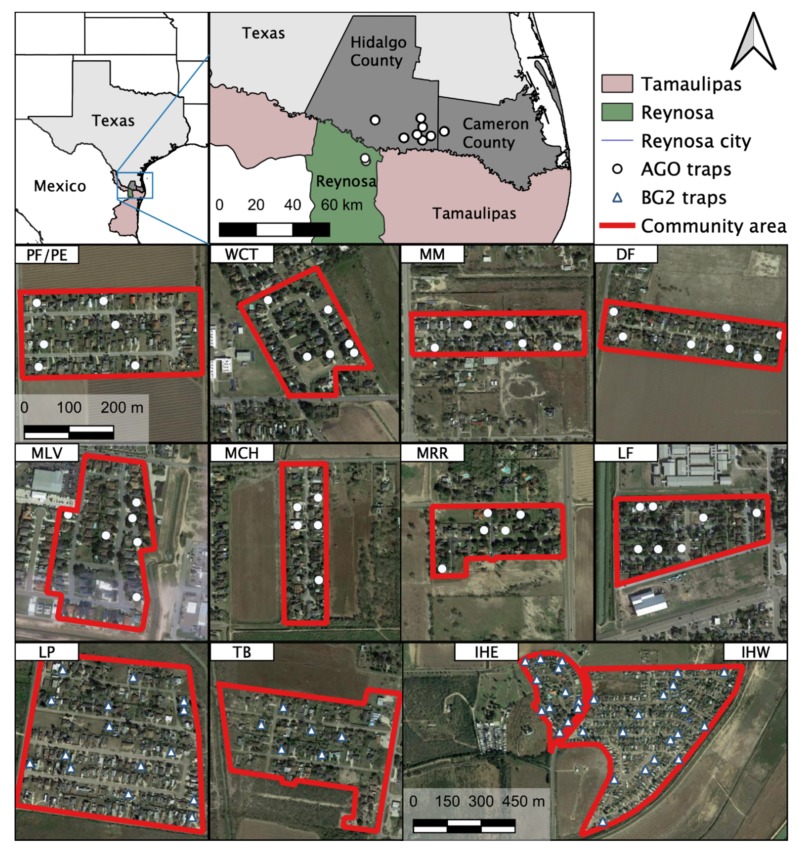
Study sites and location of traps in the Lower Rio Grande Valley (LRGV), South Texas and Reynosa (municipality), Tamaulipas. The 12 study locations in the LRGV are included as individual maps: PF/PE = Progreso Fresno/Progreso Encino; WCT = Weslaco, Christian Court; MM = Mercedes, Mesquite; DF = Donna, Figueroa; MLV = McAllen, La Vista; MCH = Mercedes, Chapa; MRR = Mercedes, Rio Rico; LF = La Feria; LP = La Piñata; TB = Tierra Bella; IHE = Indian Hills East; IHW = Indian Hills West.

**Figure 2 viruses-12-00453-f002:**
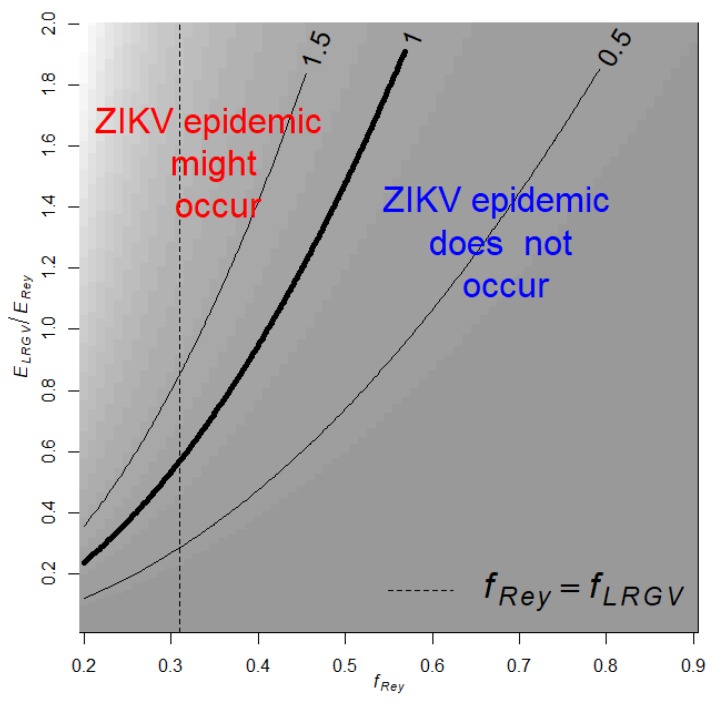
Contour plot of $R_{0}$ in the LRGV as a function on the relative risk of human exposure to *Ae. aegypti* in the LRGV compared to Reynosa ($E_{LRGV}/E_{Rey}$), and the proportion of *Ae. aegypti* feeding on humans in Reynosa (*f_Rey_*). The dashed vertical line indicates when $\frac{f_{LRGV}}{f_{Rey}}=1$. Background shading corresponds to contour predictions.

**Table 2 viruses-12-00453-t002:** Blood meal analysis results and forage ratios for *Ae. aegypti*.

Host	Count (%)	Forage Ratio (95% CI)
Dog	93 (50%)	1.61 (1.43–1.84)
Human	57 * (31%)	0.81 (0.73–0.91)
Cat	22 (12%)	0.91 (0.73–1.13)
Chicken	6 (3%)	0.19 (0.16–0.24)
Sheep	3 (1.6%)	2.69 ** (1.01–8.06)
Opossum	2 (1%)	1.19 (0.51–2.69)
Pig	3 (1.6%)	2.69 (1.01–8.06)
***Total***	186	

*—includes two mixed meals (human-dog). **—forage ratio estimated based on lowest response from vertebrate surveys (pigs).

**Table 3 viruses-12-00453-t003:** Blood meal analysis results and forage ratios for *Cx. quinquefasciatus*.

Host	Count (%)	Forage Ratio (95% CI)
Chicken	82 (67%)	3.92 (3.33–4.87)
Dog	27 (22%)	0.71 (0.63–0.81)
House sparrow	6 (5%)	-
Western kingbird	1 (0.8%)	-
Human	1 (0.8%)	0.02 (0.02–0.02)
Cat	1 (0.8%)	0.06 (0.05–0.08)
Pig	1 (0.8%)	1.36 (0.51–4.07)
Plain chachalaca	1 (0.8%)	-
Curvebilled thrasher	1 (0.8%)	-
Northern mockingbird	1 (0.8%)	-
Rock dove	1 (0.8%)	-
***Total***	123

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
