# Peer review of "High Rate of Non-Human Feeding by Aedes aegypti Reduces Zika Virus Transmission in South Texas"

_viruses, 2020, doi:10.3390/v12040453_

Round 1
Reviewer 1 Report
General comments
The Aedes aegypti mosquito is the principal vector of dengue and zika and is thought to be highly anthropophilic. Yet relatively few studies have surveyed natural mosquito populations to analyze host feeding preferences in this species. Olson et al. analyze blood meals from Aedes aegypti and Culex quinquefasciatus mosquitoes collected from BG-Sentinel and gravid ovitraps in southern Texas, USA. They find that while C. quinquefasciatus fed mainly on birds (as expected), fewer than 50% of Ae. aegypti with identifiable blood meals fed on humans, with most blood meals instead coming from dogs. This is in contrast to almost all previous studies (compiled in Table 1 here) where this species largely feeds on humans. The authors then use a combination of mosquito surveys, questionnaires, satellite data and disease data to develop a model to evaluate the potential risk of a Zika outbreak in South Texas. They conclude that high rates of feeding on non-human animals in this location reduces the risk of zika transmission and therefore outbreak potential. However given the high rates of feeding on dogs, this raises the issue of this species contributing to the transmission of enzootic pathogens.
The article is well-written and the findings are interesting, as they challenge the long-standing assumption that Ae. aegypti is highly anthropophilic. I only have a few small suggestions for improvement.
Specific comments
Section 2.1 – How specifically were species identified? E.g. identification keys.
Section 2.2 paragraph 2 - Please provide a table with sequences of all the primers used (as a supplementary table) since it is not easy to find them through the references cited. Although reference 25 lists some primers in the text, it is unclear if they are the same ones used here. It would also be worth including the primers for mosquito species identification here too.
2.2 final paragraph – Given that humans are the expected host of Ae. aegypti, has human blood also been tested as a positive control? It seems strange to have several negative controls and not a positive control.
2.10 – Equations are low resolution, making them difficult to read. This may have been an issue with the conversion process for peer review, but needs to be corrected in the final version.
2.10 paragraph 2 – Is it reasonable to assume that only these two factors differ? Couldn’t there be other reasons for a difference in R0? One example is genetic differences between mosquito populations (leading to differences in vector competence (b) and EIP).
Table 1 – The abbreviations in the Method column need a key, especially “Ab”.
Figure 2 – What does the shading indicate? There is no mention of this in the figure legend or main text.
Paragraph 4 of discussion, line 147-148. There is also some experimental evidence of successful reproduction (and even adaptation to) feeding on non-human blood in Ae. aegypti (see for example https://www.mdpi.com/2075-4450/9/4/140), providing further support for this hypothesis.
Author Response
Review 1
The Aedes aegypti mosquito is the principal vector of dengue and zika and is thought to be highly anthropophilic. Yet relatively few studies have surveyed natural mosquito populations to analyze host feeding preferences in this species. Olson et al. analyze blood meals from Aedes aegypti and Culex quinquefasciatus mosquitoes collected from BG-Sentinel and gravid ovitraps in southern Texas, USA. They find that while C. quinquefasciatus fed mainly on birds (as expected), fewer than 50% of Ae. aegypti with identifiable blood meals fed on humans, with most blood meals instead coming from dogs. This is in contrast to almost all previous studies (compiled in Table 1 here) where this species largely feeds on humans. The authors then use a combination of mosquito surveys, questionnaires, satellite data and disease data to develop a model to evaluate the potential risk of a Zika outbreak in South Texas. They conclude that high rates of feeding on non-human animals in this location reduces the risk of zika transmission and therefore outbreak potential. However given the high rates of feeding on dogs, this raises the issue of this species contributing to the transmission of enzootic pathogens.
The article is well-written and the findings are interesting, as they challenge the long-standing assumption that Ae. aegypti is highly anthropophilic. I only have a few small suggestions for improvement.
Specific comments
Section 2.1 – How specifically were species identified? E.g. identification keys.
Author response: Mosquito species were initially identified morphologically and later confirmed by molecular analysis. We added the morphological key used when processing mosquitoes in the field (Section 2.1, Lines 106-108).
Section 2.2 paragraph 2 - Please provide a table with sequences of all the primers used (as a supplementary table) since it is not easy to find them through the references cited. Although reference 25 lists some primers in the text, it is unclear if they are the same ones used here. It would also be worth including the primers for mosquito species identification here too.
Author response: Thank you for this suggestion. We added two supplemental tables that give details about the primers used in this study.
2.2 final paragraph – Given that humans are the expected host of Ae. aegypti, has human blood also been tested as a positive control? It seems strange to have several negative controls and not a positive control.
Author response: We did include positive controls in the blood meal analysis although we chose species (iguana, deer, tiger, crane, and sheep) which were unlikely to be observed as hosts for field collected specimens. Given concern for amplicon contamination, we avoided using humans as a positive control. However, our prior published work using some of these same blood meal analysis primers included human as a positive control, which amplified consistently with all primer pairs (see Hamer et al. 2009. Host Selection by Culex pipiens Mosquitoes and West Nile Virus Amplification. Am. J. Trop. Med. Hyg. 80: 268-278.).
2.10 – Equations are low resolution, making them difficult to read. This may have been an issue with the conversion process for peer review, but needs to be corrected in the final version.
Author response: In the word version of the MS, the equations appear to be sufficient resolution. We are unsure what version reviewers obtained so we will consult with editorial staff to ensure the equations, tables, and figures are of good resolution.
2.10 paragraph 2 – Is it reasonable to assume that only these two factors differ? Couldn’t there be other reasons for a difference in R0? One example is genetic differences between mosquito populations (leading to differences in vector competence (b) and EIP).
Author response: Reynosa is immediately across the border from the U.S. and our updated map shows the mosquito sampling locations in Reynosa and the LRGV, which is about 20-40km apart. Although mosquitoes are well known to have intrinsic factors that influence vector competence such as different genetics, it is unlikely that genetic substructure exists at this scale and in this region given high opportunity for gene flow across the border. However human herd immunity is likely to vary in this context and our methods section cites the Padmanadha et al. 2015 paper which documents this factor (section 2.10 line 272).
Table 1 – The abbreviations in the Method column need a key, especially “Ab”.
Author response: We added a key at the bottom of Table 1 to clarify these abbreviations.
Figure 2 – What does the shading indicate? There is no mention of this in the figure legend or main text.
Author response: The shading is the background of the contour plot which corresponds to the lines on the plot. These are comparing the relative risk of human exposure to Ae. aegypti in LRGV compared to Reynosa and the proportion of Ae. aegypti feeding on humans in Reynosa. We clarified this in the figure legend.
Paragraph 4 of discussion, line 147-148. There is also some experimental evidence of successful reproduction (and even adaptation to) feeding on non-human blood in Ae. aegypti (see for example https://www.mdpi.com/2075-4450/9/4/140), providing further support for this hypothesis.
Author response: Thank you. We added this article as another reference at line 450.
Reviewer 2 Report
In their paper untitled “High non-human feeding by Aedes aegypti reduces Zika virus transmission in South Texas » Olson et al, report results from a study carried out in residential environments of South Texas to describe host feeding patterns of Ae. aegypti and incidentally of Cx. quinquefasciatus. Mosquito collections were done in several neighborhoods in the Lower Rio Grande Valley (LRGV) along the U.S.–Mexico border (from September 2016 to December) using BG sentinel traps and Autocidal Gravid Ovitraps. When collected, blood-engorged females were analyzed using molecular tools to determine animal vs human bloodmeal sources. Unexpectedly for Ae. aegypti considered as highly anthropophilic, results indicates most bloodmeals have an animal origin (mainly dogs) while humans account only for 31% of bitten hosts. Additional analyzes taking into account estimates of host availability clearly showed a low utilization of humans compared to animal. Authors also developped an interesting R0 modeling approach using epidemiological Zika data fror Reynosa located on the other side of the USA-Mexico border. Globally this article is very informative, despite I regret host-choice experiments were not included in this study. recommend to publish it, but I have a few minor points for authors to consider for revision.
- P2, l13 : « the Asian tiger mosquito (Ae. albopictus), a secondary vector for these viruses » : Add "in USA" after "these viruses" because Ae. albopictus is the primary vector in other places of the world.
- P3, figure 1 : Place Reynosa on the map.
- P4 « bool meal analysis » : Not clear for me if bloodfed females were collected or not in Reynosa with AGO and, if yes, why these mosquitoes were not analyzed here.
- P5, l29 : Ae. aegypti formosus is not restricted to forest in Africa as it also occurs in urban areas of Africa.
- P5, l30 : Data from African studies (Nigeria, Tanzania and Kenya) are probably from Ae. aegypti formosus
Discussion
- P3, l93 : « A half century later, this observation remains the same given that only 18 studies have published Ae. aegypti host feeding » : Not true because, a lot of other studies exist but have not been considered here because because focussing on Ae. aegypti formosus.
- P3,l97 : « Zooprophylaxis is the concept that the presence of incompetent hosts can ‘waste’ bites from vector species and reduce the transmission of an infectious agent… » : Author should add something about the « dilution effect »
- P4, l44 : "The host preference of the sylvatic and domesticated forms of Ae. aegypti in east Africa is considered to have a genetic basis" : Add also McBride CS et al 2014 (Nature)
- P5, l82: « due to the abundance of non-human hosts in the residential neighborhood, the low human density, and social practices of minimizing mosquito entry indoors » : Also due to difference in human outdoor activities?
- P5,l83: « The high rate of non-human blood meals of Ae. aegypti suggests zooprophylaxis against DENV, ZIKV, and CHIKV is occurring in the LRGV » : "zooprophylaxis" is not adapted here because there is no intention for controlling human biting by Ae. aegypti. "a trophic deviation to animals" leading to a dilution effect susceptible to limit Denv, Zikv and Chikv transmission.
Author Response
Review 2
In their paper untitled “High non-human feeding by Aedes aegypti reduces Zika virus transmission in South Texas » Olson et al, report results from a study carried out in residential environments of South Texas to describe host feeding patterns of Ae. aegypti and incidentally of Cx. quinquefasciatus. Mosquito collections were done in several neighborhoods in the Lower Rio Grande Valley (LRGV) along the U.S.–Mexico border (from September 2016 to December) using BG sentinel traps and Autocidal Gravid Ovitraps. When collected, blood-engorged females were analyzed using molecular tools to determine animal vs human bloodmeal sources. Unexpectedly for Ae. aegypti considered as highly anthropophilic, results indicates most bloodmeals have an animal origin (mainly dogs) while humans account only for 31% of bitten hosts. Additional analyzes taking into account estimates of host availability clearly showed a low utilization of humans compared to animal. Authors also developped an interesting R0 modeling approach using epidemiological Zika data fror Reynosa located on the other side of the USA-Mexico border. Globally this article is very informative, despite I regret host-choice experiments were not included in this study. recommend to publish it, but I have a few minor points for authors to consider for revision.
- P2, l13 : « the Asian tiger mosquito (Ae. albopictus), a secondary vector for these viruses » : Add "in USA" after "these viruses" because Ae. albopictus is the primary vector in other places of the world.
Author response: Ae. albopictus is the primary vector in a few locations of the world but this is relatively rare compared to Ae. aegypti as the dominant vector. Perhaps the following is a revision that acknowledges this:
“Despite this wide distribution of the primary vector and the Asian tiger mosquito (Ae. albopictus), a secondary vector for these viruses in many locations, the only regions experiencing autochthonous transmission of DENV, CHIKV, and ZIKV by mosquito exposure are South Florida and South Texas [9,10].”
We added the phrase “in many locations” in section 1 at line 55 to better clarify this.
- P3, figure 1 : Place Reynosa on the map.
Author response: Thanks for this observation as we have now corrected Figure 1 to show Reynosa, Tamaulipas as well as the AGO sampling locations in the city of Reynosa.
- P4 « bool meal analysis » : Not clear for me if bloodfed females were collected or not in Reynosa with AGO and, if yes, why these mosquitoes were not analyzed here.
Author response: Our internal funding to conduct the blood meal analysis was only for the bloodfeds collected from South Texas. Therefore, we did not process the bloodfeds collected from Reynosa. We clarified this in the methods (section 2.1, line 91) and an additional statement was added in the discussion identifying this as high priority for future research (section 4, line 427).
- P5, l29 : Ae. aegypti formosus is not restricted to forest in Africa as it also occurs in urban areas of Africa.
Author response: This is a good point so we have removed the words “and sylvatic” (line 449) to avoid implying that this species is restricted to forests.
- P5, l30 : Data from African studies (Nigeria, Tanzania and Kenya) are probably from Ae. aegypti formosus
Author response: Related to the previous response, we changed “sylvatic” to “ancestral” on line 450. The Nigeria and Tanzania studies were conducted prior to 1957 when Mattingly provisionally named the three subspecies of Ae. aegypti based upon coloration (see Macdonald, 1967), so it’s difficult to know what subspecies they collected. The mosquitoes are simply referred to as Ae. aegypti. The Kenya study by Heisch reported their mosquitoes as Ae. aegypti queenslandensis.
Discussion
- P3, l93 : « A half century later, this observation remains the same given that only 18 studies have published Ae. aegypti host feeding » : Not true because, a lot of other studies exist but have not been considered here because because focussing on Ae. aegypti formosus.
Author response: We found a total of 21 studies that published Ae. aegypti host-feeding results, but removed 3 studies that were clearly analyzing Ae. aegypti formosus from Table 1 and our analyses. We changed this statement in the discussion from “18” to “21” to include the studies excluded.
- P3,l97 : « Zooprophylaxis is the concept that the presence of incompetent hosts can ‘waste’ bites from vector species and reduce the transmission of an infectious agent… » : Author should add something about the « dilution effect »
Author response: This is a good point that variation in the community of hosts can influence the disease risk. For mosquito-borne viruses, most studies suggest the diversity and richness of vertebrates is less important than the ‘host community competence’. Therefore, we have added the following sentence at line 398:
“Furthermore, prior studies have identified that arboviral transmission potential is impacted by host community composition and competence (Hamer et al. 2011, Koolhof and Carver 2017). For human-amplified urban arboviruses like ZIKV, less feeding on humans and more feeding on non-competent hosts (vertebrates with a low duration and magnitude of viremia unable to re-infect Ae. aegypti), will dilute virus transmission.
- P4, l44 : "The host preference of the sylvatic and domesticated forms of Ae. aegypti in east Africa is considered to have a genetic basis" : Add also McBride CS et al 2014 (Nature)
Author response: Thanks for the suggested study which we have added as a citation.
- P5, l82: « due to the abundance of non-human hosts in the residential neighborhood, the low human density, and social practices of minimizing mosquito entry indoors » : Also due to difference in human outdoor activities?
Author response: We did not quantify human behavior among these two communities although the potential for more time spent outdoors is certainly related to the risk of human exposure to Ae. aegypti. We modified this sentence to end with ‘and social practices of minimizing risk of exposure to Ae. aegypti’ which captures both indoor and outdoor feeding (section 5, line 491).
- P5,l83: « The high rate of non-human blood meals of Ae. aegypti suggests zooprophylaxis against DENV, ZIKV, and CHIKV is occurring in the LRGV » : "zooprophylaxis" is not adapted here because there is no intention for controlling human biting by Ae. aegypti. "a trophic deviation to animals" leading to a dilution effect susceptible to limit Denv, Zikv and Chikv transmission.
Author response: We agree that people did not add dogs to their communities for the purpose of wasting bites from Ae. aegypti. We have modified this concluding sentence by removing the term ‘zooprophylaxis’.
Reviewer 3 Report
In this study, Olson et al. identify underutilization of human hosts by Aedes aegypti mosquitoes as a major factor that accounts for the low levels of human Zika cases in Lower Rio Grande Valley, South Texas, US, compared to Tamaulipas State, Northern Mexico, in recent years. This underutilization can be primarily explained by low risk of human exposure to mosquito bites which drives feeding on alternative, abundant non-human vertebrate hosts. This finding complements other recent studies in the area on socio-economical factors and informs disease control and management policies.
As the acknowledgement section suggests, this manuscript has been reviewed previously and I do not have any major criticisms to add. Overall, the study design, presentation of data and conclusions are acceptable in their current format, and the manuscript is well written. Minor comments below.
Out of scientific interest, I would like to know if the correlation between low human feeding rates and high abundance of alternative non-human vertebrate hosts described in this study is replicated in other geographic locations presented in Table 1. Reversely, are there simply less alternative hosts present in countries such as India (reference 57) which would explain why humans are the preferred host? Or would the authors suspect that factors such as housing standards are accounting for the differences? I understand it would be difficult to normalise human host usage to non-human vertebrate host availability for other studies as these data will not be readily available. But if the authors could present an example it would help decide if what they have found is a general trend or specific to their local mosquito populations.
Minor comments:
- Title should be changed to "High rates of non-human...".
- In the abstract geographic locations are mentioned without any context. I had to find a map to clarify where these are. This could be simplified to South Texas vs North Mexico. More details can be given in the introduction.
- Methods 2.10: The formulae are very difficult to read in the pdf version. There is one long paragraph between formulae 2 and 3 (there were no line numbers, sorry) that could be placed in the introduction instead.
- Figure S2: x-axes should be "Time (weeks)" to make it easier for the reader to understand the figure.
- Results 3.1: I would have liked to have a sentence of explanation as to why Cx. quinquefasciatus were collected and analysed given that they do not play a major role in ZIKV transmission.
Author Response
Review 3
In this study, Olson et al. identify underutilization of human hosts by Aedes aegypti mosquitoes as a major factor that accounts for the low levels of human Zika cases in Lower Rio Grande Valley, South Texas, US, compared to Tamaulipas State, Northern Mexico, in recent years. This underutilization can be primarily explained by low risk of human exposure to mosquito bites which drives feeding on alternative, abundant non-human vertebrate hosts. This finding complements other recent studies in the area on socio-economical factors and informs disease control and management policies.
As the acknowledgement section suggests, this manuscript has been reviewed previously and I do not have any major criticisms to add. Overall, the study design, presentation of data and conclusions are acceptable in their current format, and the manuscript is well written. Minor comments below.
Out of scientific interest, I would like to know if the correlation between low human feeding rates and high abundance of alternative non-human vertebrate hosts described in this study is replicated in other geographic locations presented in Table 1. Reversely, are there simply less alternative hosts present in countries such as India (reference 57) which would explain why humans are the preferred host? Or would the authors suspect that factors such as housing standards are accounting for the differences? I understand it would be difficult to normalise human host usage to non-human vertebrate host availability for other studies as these data will not be readily available. But if the authors could present an example it would help decide if what they have found is a general trend or specific to their local mosquito populations.
Minor comments:
- Title should be changed to "High rates of non-human...".
Author response: Revised as suggested.
- In the abstract geographic locations are mentioned without any context. I had to find a map to clarify where these are. This could be simplified to South Texas vs North Mexico. More details can be given in the introduction.
Author response: We changed the abstract to focus on South Texas and Northern Mexico which can then be elaborated on elsewhere in the manuscript.
- Methods 2.10: The formulae are very difficult to read in the pdf version. There is one long paragraph between formulae 2 and 3 (there were no line numbers, sorry) that could be placed in the introduction instead.
Author response: Our response to the same comment from a different reviewer is here: “In the word version of the MS, the equations appear to be sufficient resolution. We are unsure what version reviewers obtained so we will consult with editorial staff to ensure the equations, tables, and figures are of good resolution.”
- Figure S2: x-axes should be "Time (weeks)" to make it easier for the reader to understand the figure.
Author response: Thank you for this suggestion. We updated Figure S2 as you suggested.
- Results 3.1: I would have liked to have a sentence of explanation as to why Cx. quinquefasciatus were collected and analysed given that they do not play a major role in ZIKV transmission.
Author response: We added the following sentence to the discussion which provides an explanation as to why our preference is to present the Culex and Aedes blood meal results together (Section 4, line 470).
“Both the bloodfed Culex and Aedes were processed with the exact same protocol and the contrasting results provides more confidence in the accuracy of the identified blood meals.”